# Masked Image Modeling with Denoising Contrast

**Kun Yi**[1][*]  **Yixiao Ge**[1][*][†]  **Xiaotong Li**[1,4]  **Shusheng Yang**[1,5]
**Dian Li**[3]  **Jianping Wu**[6]  **Ying Shan**[1]  **Xiaohu Qie**[2]

[1]ARC Lab, [2]Tencent PCG    [3]Foundation Technology Center, [2]Tencent PCG
[4]Peking University   [5]Huazhong University of Science and Technology    [6]Tsinghua University

[*]**equal contribution**    [†]**corresponding author**

## Abstract

Since the development of self-supervised visual representation learning from contrastive learning to masked image modeling (MIM), there is no significant difference in essence, that is, how to design proper pretext tasks for vision dictionary look-up. MIM recently dominates this line of research with state-of-the-art performance on vision Transformers (ViTs), where the core is to enhance the patch-level visual context capturing of the network via denoising auto-encoding mechanism. Rather than tailoring image tokenizers with extra training stages as in previous works, we unleash the great potential of contrastive learning on denoising auto-encoding and introduce a pure MIM method, ConMIM, to produce simple intra-image inter-patch contrastive constraints as the sole learning objectives for masked patch prediction. We further strengthen the denoising mechanism with asymmetric designs, including image perturbations and model progress rates, to improve the network pre-training. ConMIM-pretrained models with various scales achieve competitive results on downstream image classification, semantic segmentation, object detection, and instance segmentation tasks, *e.g.*, on ImageNet-1K classification, we achieve 83.9% top-1 accuracy with ViT-Small and 85.3% with ViT-Base without extra data for pre-training. Code will be available at `https://github.com/TencentARC/ConMIM`.

## 1 Introduction

The great success of self-supervised learning in natural language processing (NLP) tasks, *e.g.*, BERT (Devlin et al., 2019) and GPT (Radford et al., 2018; 2019), has sparked several revolutions in visual representation learning, during which the development of *vision dictionary look-up* is the most critical. In the age of convolutional neural networks (CNNs) (He et al., 2016; Krizhevsky et al., 2012), prominent works (He et al., 2020; Chen et al., 2020) perform self-supervised learning with a pretext task of *instance-level dictionary look-up via contrastive learning* as demonstrated in Figure 1(a). With the advent of vision Transformers (ViTs) (Dosovitskiy et al., 2021), the gap between vision and NLP tasks has been further narrowed since the introduction of *patch-level dictionary look-up via masked image modeling* in a pioneer work BEiT (Bao et al., 2022) (see Figure 1(b)).

The introduction of masked image modeling (Bao et al., 2022), inspired by masked language modeling (Devlin et al., 2019) in NLP tasks, ushers in a new fad for self-supervised learning using vision Transformers (Dosovitskiy et al., 2021), *i.e.*, a portion of vision tokens are randomly masked and then recovered by the Transformer network being trained. Concurrent works (Dong et al., 2021; Li et al., 2022; Wei et al., 2022) make efforts to design patch-level dictionaries, image tokenizers in other words, to build proper learning objectives (*i.e.*, vision token ids) for masked image modeling. Though advanced results can be achieved, the off-the-shelf image tokenizers, *e.g.*, discrete VAE (Ramesh et al., 2021) used in BEiT (Bao et al., 2022), depend on extra training stages and data knowledge, rendering an inflexible two-stage pre-training paradigm.

We would like to call for a revisit of the superiority of masked image modeling over contrastive learning on self-supervised learning with vision Transformers. Since they are essentially both designed

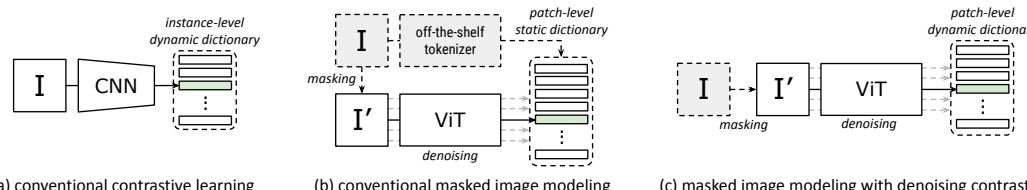

Figure 1: Conventional contrastive learning methods (*e.g.*, MoCo (He et al., 2020), SimCLR (Chen et al., 2020)) and masked image modeling methods (*e.g.*, BEiT (Bao et al., 2022), PeCo (Dong et al., 2021)) both perform the pretext task of vision dictionary look-up, where the superiority of the latter ones lie in the patch-level denoising auto-encoding mechanism to enable fine-grained visual context understanding of vision Transformers (Dosovitskiy et al., 2021). We introduce to cast masked image modeling as denoising contrastive learning to avoid the extra training stages of image tokenizer, rendering a flexible, simple and effective pre-training paradigm.

towards vision dictionary look-up, the key difference lies in the patch-level denoising auto-encoding mechanism in masked image modeling, which encourages the network's capability to capture fine-grained visual context and semantics. As for the auto-encoding objective, we do not have to intentionally discretize the continuous visual signals as words in NLP tasks to cast the masked prediction as a classification task. Instead, we can give full play to the wisdom of contrastive learning, which has good capability to structure the visual space with semantically meaningful representations. To this end, we introduce a new pre-training method for masked image modeling, namely, ConMIM, to get rid of extra tokenizing networks by revitalizing contrastive learning, as shown in Figure 1(c).

Our ConMIM casts masked patch prediction in self-supervised image pre-training as denoising contrastive learning. The corrupted input with a large proportion of patches masked is fed into the encoder, a plain vision Transformer in general. The encoder learns to recover the representations of the masked patches, which are predicted by feeding the full input into the encoder. The training objective is formed by an intra-image inter-patch contrastive loss. To be specific, patch representations of a full input image build a dynamic dictionary, and patches from the same positions as the masked ones of the corrupted input serve as their positive keys, respectively. The remaining ones from different positions but in a same image are the negative keys. To further improve the network via a stronger denoising auto-encoding mechanism, we introduce asymmetric designs in ConMIM training, including asymmetric image perturbations and asymmetric model progress rates. We adopt a strong augmentation for the full input while a weak augmentation for the corrupted input. For the image encoder, the slowly progressing momentum encoder (He et al., 2020) is employed for the full input to embed more challenging but semantically consistent learning targets.

We perform self-supervised learning with ConMIM on ImageNet (Deng et al., 2009), and then fine-tune the pre-trained vision Transformers with various scales on image classification, semantic segmentation, object detection and instance segmentation. Unlike those employ large models with super-scale extra data knowledge, ConMIM excels especially at small-scale architectures, which render a more challenging task for effective pre-training as well as a more practical task in real-world applications. With a vanilla ViT-Small model, we achieve 83.9% top-1 accuracy using only ImageNet-1K, suggesting that useful knowledge is exploited from data. This significantly outperforms the baseline BEiT (Bao et al., 2022) and the comparable MIM methods without tokenizers (*e.g.*, MAE (He et al., 2022), iBOT (Zhou et al., 2022)) due to the stronger semantic structured regularization in ConMIM. Other than the promising results, we would like to draw public attention to unleash the great potential of "outdated" contrastive learning in visual representation learning.

## 2 RELATED WORK

**Self-supervised learning via vision dictionary look-up.** The pretext task of contrastive learning (Chen et al., 2020; He et al., 2020; Caron et al., 2020) dominates self-supervised visual pre-training in the era of CNNs. Contrastive learning methods generally perform *instance-level dictionary look-up*. The anchors are pulled closer to their positive keys at the same time pushing away from the negative keys. The establishment of vision dictionaries is critical for the contrast regularization. For example, the seminal work MoCo (He et al., 2020) builds the vision dictionary with a first-in-first-out queue, driven by a momentum encoder. The concurrent work SimCLR (Chen et al.,

2020) uses a large batch size to enlarge the dictionary with more negative keys. SwAV (Caron et al., 2020) further introduces an online clustering algorithm in an unsupervised manner, and the cluster assignments serve for the dictionary keys. Despite the great achievements with CNNs, these methods are gradually abandoned with the introduction of ViTs (Dosovitskiy et al., 2021) due to the lack of inductive bias, which requires stronger supervision for better pre-training performance.

Researchers attempt to reproduce the success of masked language modeling (Devlin et al., 2019) in self-supervised learning of ViTs via *patch-level dictionary look-up*. Specifically, BEiT (Bao et al., 2022) introduces a new pretext task, namely, masked image modeling, for visual pre-training. They tokenize high-dimensional images into discrete vision tokens by a discrete VAE (Ramesh et al., 2021) to establish a static patch-level dictionary as in NLP tasks. A proportion of image patches are randomly masked, the backbone network is then trained to recover the vision token ids of the masked patches, rendering a denoising mechanism. Follow-up works make efforts to further improve the static dictionaries, *e.g.*, mc-BEiT (Li et al., 2022) introduces eased and refined dictionaries with multiple choices. PeCo (Dong et al., 2021) proposes to produce perceptual-aware keys in the patch-level dictionary. Though promising results, these methods all require extra training stages and even extra data for obtaining a proper image tokenizer.

We would also like to mention and thank the classic denoising auto-encoding methods (Vincent et al., 2010; Seung, 1997). Though they did not mask patches in Transformer, these pilot works on auto-encoding and emphasized reconstruction have well inspired the deep learning community.

**Tokenizer-free masked image modeling (MIM) methods.** There are other recent works that cast MIM as a pixel-level reconstruction task (*e.g.*, MAE (He et al., 2022)) or a self-distillation task (*e.g.*, iBOT (Zhou et al., 2022)) rather than dictionary look-up. However, they fail to achieve competitive results using the same training epochs and perform especially unsatisfactorily on small-scale architectures due to the weak regression constraints (see Appendix B.4). Moreover, iBOT is not a pure MIM method as it heavily depends on the vanilla DINO (Caron et al., 2021) loss (*i.e.*, the global self-distillation loss on [CLS] tokens). It actually conducts MIM on top of DINO and argues that MIM alone hardly captures visual semantics. However, we would like to clarify that it is actually due to the improper MIM constraints. Contrastive learning is proven to be good at structuring the visual space but does not been successfully employed in MIM before. We propose a flexible pure MIM method without extra dependencies, including offline tokenizer or global discrimination loss.

**Dense contrast vs. denoising contrast.** There are some previous works on contrastive learning devoted to taking local feature representations into consideration, *e.g.*, DenseCL (Wang et al., 2021). Though the form of InfoNCE (Van den Oord et al., 2018) is similar, they show significant differences from our ConMIM in both motivation and method design. They focus on how to learn better pre-trained weights for dense downstream tasks, *e.g.*, object detection and segmentation, but hardly encourage the patch-level visual context reasoning as it is a contrastive-only task, showing inferior performance on ViT pre-training. Moreover, DenseCL depends on the global discrimination loss to ensure correct local correspondences and needs to carefully balance the global and local constraints. Such a chicken-and-egg problem can be seamlessly addressed in our well-designed denoising mechanism, including both the masking operation and the asymmetric designs. See Appendix B.4.1 for experimental comparisons. There are also some concurrent works (Tao et al., 2022; Huang et al., 2022) that study contrastive learning in MIM. ConMIM is conducted independently with them.

## 3 PRELIMINARIES

The pretraining-and-then-finetuning paradigm has been proven to be effective for visual representation learning and various downstream tasks, where self-supervised pre-training is the most popular. Since there are no ground-truth annotations available, the design of pretext tasks is critical to the pre-training performance. Though driven by various motivations and progressing architectures (He et al., 2016; Dosovitskiy et al., 2021), the pretext task of visual self-supervised learning is essentially to perform vision dictionary look-up, inspired by the success of NLP tasks.

### 3.1 CONTRASTIVE LEARNING: INSTANCE-LEVEL VISION DICTIONARY LOOK-UP

From the perspective of vision dictionary look-up, prominent contrastive learning methods establish instance-level vision dictionaries via a fixed-length queue (He et al., 2020) or batch-wise samples (Chen et al., 2020). The keys in the dictionary are dynamically updated as pre-training proceeds.

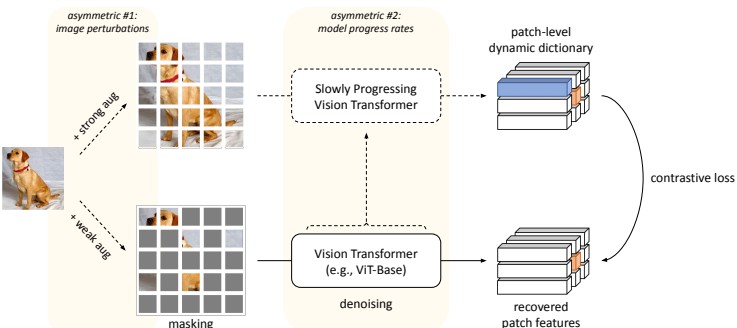

Figure 2: Our ConMIM performs the masked patch prediction with denoising contrast, coupling with two asymmetric designs to achieve state-of-the-art performance on self-supervised image pre-training. The slowly progressing vision Transformer is a snapshot of the backbone network under training, and we do not require any off-the-shelf image tokenizers. The training objective of denoising contrastive loss performs a patch-level look-up from dynamic vision dictionaries and enhances the network's capability to capture more fine-grained visual context.

Given an image $x$, its feature representation is encoded by feeding it into the backbone network, *i.e.*, $f(x)$. An InfoNCE loss (Van den Oord et al., 2018) is employed to regularize this representation, bringing it closer to its positive key $k_+$ while staying away from negative keys, denoted as

$$\mathcal{L}_{\text{con}}(x) = -\log \frac{\exp\left(\langle f(x), k_+ \rangle / \tau\right)}{\sum_{i=1}^{K} \exp\left(\langle f(x), k_i \rangle / \tau\right)}, \tag{1}$$

where $\langle \cdot, \cdot \rangle$ is the cosine similarity measured by the dot product between two $L_2$-normalized features, $\tau$ is the temperature hyper-parameter, $k$ is the dynamic key, and $K$ is the dictionary size. Generally, the positive key is built by another view of the same instance (Tian et al., 2020), *e.g.*, different image augmentations.

## 3.2 MASKED IMAGE MODELING: PATCH-LEVEL VISION DICTIONARY LOOK-UP

With the popularity of Transformer architectures (Dosovitskiy et al., 2021) in computer vision tasks, the pretext task of masked image modeling gradually dominates visual representation learning. It randomly masks a large percentage of image patches and trains the backbone network to recover the token ids of corrupted image via more fine-grained patch-level vision dictionary look-up. The dictionary is generally static and pre-defined by an off-the-shelf image tokenizer (Ramesh et al., 2021; Esser et al., 2021), which converts continuous visual signals into discrete keys. For example, in the seminal work BEiT (Bao et al., 2022), a pre-learned discrete VAE is adopted as the tokenizer. The masked patch prediction is then cast as a classification task with cross-entropy loss,

$$\mathcal{L}_{\text{mim}}(x) = \mathbb{E}_{j \in \mathcal{M}} \left[ -\log p(y_j | f(\hat{x})_j) \right], \tag{2}$$

where $\mathcal{M}$ denotes the set of masked patch indices, $\hat{x}$ is the corrupted image after randomly masking, $y_j$ is the positive key index in the patch-level dictionary, and $p(\cdot | \cdot)$ indicates the probability that correctly identifies the recovered patch $f(\hat{x})_j$ with a patch index of $j$.

## 4 MASKED IMAGE MODELING WITH DENOISING CONTRAST

Despite that existing contrastive learning and masked image modeling methods optimize the backbone network towards different training objectives (*i.e.*, InfoNCE loss and cross-entropy loss), they both attempt to learn discriminative visual representations via dictionary look-up. Two key factors lead to the state-of-the-art performance of masked image modeling. (1) More fine-grained supervision from instance-level to patch-level benefits the vision Transformer architecture known for its data-hungry properties. (2) The denoising auto-encoding mechanism, formed by the masking-and-then-predicting paradigm, encourages the capability of the backbone network to capture contextualized representations. Though promising results are achieved by existing masked image modeling methods (Bao et al., 2022; Li et al., 2022; He et al., 2022), they either require extra training stages to establish static vision dictionaries with offline image tokenizers or lack powerful MIM constraints.

To this end, we call for a revitalization of contrastive learning, which has good capability to structure the latent space for self-supervised representation learning. A new self-supervised pre-training method, ConMIM, is introduced to perform pure masked image modeling with denoising contrastive objectives while eliminating the dependence on pre-learned image tokenizers, as shown in Figure 2.

**Patch-level dynamic dictionary.** We build dynamic patch-level dictionaries to form the learning targets for masked patch prediction on-the-fly. Specifically, during each training iteration, the full input image $x$ is fed into the backbone network to embed the patch feature representations, serving as keys in the dynamic dictionary, *i.e.*, $\{f(x)_i|_{i=1}^{K}\}$ where $i$ is the patch index, $K$ is the dictionary size as well as the total number of patches within an image (*e.g.*, $K = 196$ keys for a $224 \times 224$ image with a patch size of $16 \times 16$). Without the loss of representation discriminativeness, we build separate dictionaries for each image, that is, only operate patch-level dictionary look-up within each image. We discuss this design in Sec. 5.4.2 with ablation studies using a larger or smaller dictionary size, where inferior results are achieved and require extra computational overhead.

**Denoising contrastive objective.** The corrupted image, $\hat{x}$, is then fed into the backbone network, and we denote the encoded patch feature representation of a certain masked patch as $f(\hat{x})_j, j \in \mathcal{M}$. The backbone network is trained to denoise the corrupted image and recover the masked patches through visual context reasoning. The masked patch recovery is regularized by a patch-level dictionary look-up in the form of an InfoNCE loss (Van den Oord et al., 2018),

$$\mathcal{L}_{\text{conmim}}(x) = \mathbb{E}_{j \in \mathcal{M}} \left[ -\log \frac{\exp\left(\langle f(\hat{x})_j, \text{sg}[f(x)_j]\rangle / \tau\right)}{\sum_{i=1}^{K} \exp\left(\langle f(\hat{x})_j, \text{sg}[f(x)_i]\rangle / \tau\right)} \right], \tag{3}$$

where $\text{sg}[\cdot]$ indicates stop-gradient operation. We only backpropagate the gradients of the corrupted inputs $f(\hat{x})$ because backpropagating the gradients of the full input $f(x)$ may lead to information leakage and useless denoising. With the above training objectives, the backbone network is encouraged to better capture the visual context and learns to encode local discriminative representations.

**Asymmetric design.** As patches (*e.g.*, $16 \times 16$) are small-scale inputs with less useful information and highly redundant semantics, we need to make the pre-training task more challenging to improve the backbone network. Towards this goal, the recent work MAE (He et al., 2022) proposes to mask a large proportion of patches. In our work, besides the large proportion of patch dropout, we further introduce two asymmetric designs to enable a stronger denoising regularization during pre-training.

*(1) Asymmetric image perturbations.* We adopt different data augmentations for the full input image $x$ and the corrupted image $\hat{x}$ before feeding into the backbone network. To be specific, we utilize stronger augmentations for the full input image referring to contrastive learning methods (Chen et al., 2020), including random flip, resize, crop, color distort, Gaussian blur, and solarization. And we only use basic augmentations for the corrupted image referring to masked image modeling methods (Bao et al., 2022), including random flip, resize, and crop. Note that we use the same flip, resize and crop operations for paired inputs in order to keep the patch positions consistent. We discuss the alternative options in Sec. 5.4.3. We observe that it is sub-optimal to use strong augmentations for corrupted images, and the reason might be the too difficult pretext task to regularize, *i.e.*, the backbone network needs to recover the strong-augmented corrupted input towards the full input targets with asymmetric perturbations.

*(2) Asymmetric model progress rates.* We employ different model progress rates of the backbone network for embedding the corrupted input and full input to avoid information leakage. We use the in-training network *i.e.*, the one optimized by loss backpropagation for the corrupted input while using its momentum updated version for the full input. The momentum encoder (He et al., 2020) is known as a slowly progressing model that can encode more challenging but semantically consistent key feature representations for building dictionaries. Specifically, we denote the model parameters of the backbone $f(\cdot)$ as $\theta$ and the model parameters of the momentum updated one as $\tilde{\theta}$. The momentum encoder is updated via $\tilde{\theta} = (1 - \alpha)\theta + \alpha\tilde{\theta}$ in each iteration, where $\alpha \in [0, 1]$ is the momentum coefficient. Larger coefficients indicate slower model progress.

**Pre-training setup.** ConMIM pre-training is conducted on the training set of ImageNet-1K (Deng et al., 2009) dataset in a self-supervised manner. We utilize ViT-S/16, ViT-B/16 and ViT-L/16 (Dosovitskiy et al., 2021) as the backbone networks. Following MoCo v3 (Chen* et al., 2021), we use a

| Method | Arch. | #Epochs | Acc. |
|---|---|---|---|
| scratch | ViT-B/16 | - | 81.8 |
| MoCo v3 | ViT-B/16 | 600 | 83.2 |
| DINO | ViT-B/16 | 1600 | 82.8 |
| BEiT | ViT-B/16 | 300 | 82.9 |
| ConMIM | ViT-B/16 | 300 | 83.5 |
| iBOT | ViT-B/16 | 600 | 82.0 |
| BEiT | ViT-B/16 | 800 | 83.2 |
| MAE | ViT-B/16 | 800 | 83.3 |
| ConMIM | ViT-B/16 | 800 | 83.7 |
| SimMIM | ViT-B/16 | 800 | 83.8 |
| MAE | ViT-B/16 | 1600 | 83.6 |
| iBOT | ViT-B/16 | 1600 | 84.0 |
| scratch[†] | ViT-B/16 | - | 83.1 |
| BEiT[†] | ViT-B/16 | 800 | 84.6 |
| ConMIM[†] | ViT-B/16 | 800 | **85.3** |
| MAE[†] | ViT-B/16 | 1600 | 84.9 |
| iBOT[†] | ViT-B/16 | 1600 | 85.0 |

Table 1: Top-1 accuracy (%) on ImageNet-1K (Deng et al., 2009) classification using ViT-Base (Dosovitskiy et al., 2021). All the models use only ImageNet-1K with a resolution of $224^2$ for pre-training. ([†]) The resolution of $384^2$ is used for fine-tuning. We follow the same $384^2$ fine-tuning setup of BEiT (Bao et al., 2022), *i.e.*, after fine-tuning with $224^2$, another 10% epochs are further used for fine-tuning on $384^2$.

| Method | Arch. | #Epochs | Acc. |
|---|---|---|---|
| scratch | ViT-S/16 | - | 79.8 |
| MoCo v3 | ViT-S/16 | 600 | 81.4 |
| DINO | ViT-S/16 | 1600 | 81.5 |
| BEiT | ViT-S/16 | 300 | 81.3 |
| ConMIM | ViT-S/16 | 300 | 82.0 |
| iBOT | ViT-S/16 | 600 | 81.5 |
| SimMIM | ViT-S/16 | 800 | 80.1 |
| MAE | ViT-S/16 | 800 | 80.9 |
| iBOT | ViT-S/16 | 3200 | 82.3 |
| ConMIM[†] | ViT-S/16 | 300 | **83.9** |

Table 2: Top-1 accuracy (%) on ImageNet-1K classification using ViT-Small.

| Method | Arch. | #Epochs | Acc. |
|---|---|---|---|
| scratch | ViT-L/16 | - | 82.6 |
| MoCo v3 | ViT-L/16 | 600 | 84.1 |
| MAE | ViT-L/16 | 400 | 84.3 |
| ConMIM | ViT-L/16 | 400 | 84.6 |
| BEiT | ViT-L/16 | 800 | 85.2 |
| MAE | ViT-L/16 | 800 | 85.2 |
| ConMIM | ViT-L/16 | 800 | 85.2 |
| iBOT | ViT-L/16 | 1000 | 84.8 |
| ConMIM | ViT-L/16 | 1600 | 85.5 |
| MAE | ViT-L/16 | 1600 | 85.9 |
| BEiT[†] | ViT-L/16 | 800 | 86.3 |
| ConMIM[†] | ViT-L/16 | 800 | 86.3 |
| ConMIM[†] | ViT-L/16 | 1600 | **86.5** |

Table 3: Top-1 accuracy (%) on ImageNet-1K classification using ViT-Large.

3-layer projection head on top of the backbone network for pre-training and discard it when transferring to downstream tasks. The input images are all resized to $224 \times 224$ and the patch size is set to $16 \times 16$. We follow the masking strategy of MAE (He et al., 2022), *i.e.*, 75% patches are randomly masked. The learning rate is set to 5e-4, with a warmup of 10 epochs, and cosine learning rate decay. The temperature $\tau$ is set to 0.1 and the momentum coefficient $\alpha$ is initially set to 0.996 with a cosine scheduler. ViT-B/16 and ViT-L/16 are pre-trained for 800 epochs in total and ViT-S/16 is pre-trained for 300 epochs if not specified. The other hyper-parameters are mostly the same as BEiT (Bao et al., 2022). More implementation details can be found in Appendix A.

## 5 EXPERIMENTS

We evaluate the models pre-trained by our ConMIM on different downstream tasks, including image classification on ImageNet-1K (Deng et al., 2009) (Sec. 5.1), semantic segmentation on ADE20K (Zhou et al., 2017) (Sec. 5.2), object detection and instance segmentation on COCO (Lin et al., 2014) (Sec. 5.3). We further discuss the key components in ConMIM pre-training via ablation studies in Sec. 5.4, the scalability on larger dataset in Sec. 5.5, and the running time in Sec. 5.6. Hyper-parameter details can be found in Appendix A.1 and analysis in Appendix B.1.

### 5.1 IMAGE CLASSIFICATION

We test our ConMIM by fine-tuning the pre-trained models on ImageNet-1K (Deng et al., 2009) classification, which contains 1.3M images out of 1K classes in total. We mostly follow the fine-tuning setup of BEiT (Bao et al., 2022). To be specific, we use 100 epochs with a warm-up of 20 epochs, and a layer decay of 0.65 for ViT-Base fine-tuning. 200 epochs with a warm-up of 5 epochs and a layer decay of 0.8 for ViT-Small fine-tuning. 50 epochs with a warm-up of 5 epochs and a layer decay of 0.75 for ViT-Large fine-tuning. We illustrate the evaluation results in Tables 1,2,3. We observe that models with pre-trained weights overwhelmingly outperform the ones trained from scratch by DeiT (Touvron et al., 2021), demonstrating the significance of self-supervised visual representation learning. Compared to the pioneering work of masked image modeling, BEiT (Bao et al., 2022), we consistently outperform it when fine-tuning on ViT-Base and ViT-Small models with both image resolutions of $224 \times 224$ and $384 \times 384$. Moreover, we do not require any extra

| Methods | Arch. | #Ep. | mIOU |
|---------|-------|------|------|
| BEiT[‡] | ViT-S/16 | 300 | 46.3 |
| ConMIM[‡] | ViT-S/16 | 300 | **47.7** |
| scratch | ViT-B/16 | - | 45.3 |
| MoCo v3 | ViT-B/16 | 600 | 47.2 |
| BEiT | ViT-B/16 | 800 | 45.6 |
| ConMIM | ViT-B/16 | 800 | 46.0 |
| DINO | ViT-B/16 | 1600 | 46.8 |
| MAE | ViT-B/16 | 1600 | 48.1 |
| iBOT | ViT-B/16 | 1600 | 50.0 |
| BEiT[‡] | ViT-B/16 | 800 | 47.7 |
| ConMIM[‡] | ViT-B/16 | 800 | **49.8** |
| MAE[‡] | ViT-B/16 | 1600 | 48.0 |
| scratch | ViT-L/16 | - | 49.9 |
| MoCo v3 | ViT-L/16 | 600 | 49.1 |
| BEiT[‡] | ViT-L/16 | 800 | 53.3 |
| ConMIM[‡] | ViT-L/16 | 800 | **53.7** |
| MAE | ViT-L/16 | 1600 | 53.6 |

Table 4: Semantic segmentation on ADE20K (Zhou et al., 2017) in terms of mIOU (%). "#Ep." means the number of pre-trained epochs. ([‡]) Transferring after intermediate fine-tuning on ImageNet-1K (Deng et al., 2009), which is a common practice of BERT (Devlin et al., 2019).

| Methods | Arch. | #Ep. | AP$^{box}$ | AP$^{mask}$ |
|---------|-------|------|------------|-------------|
| scratch | ViT-S/16 | - | 43.1 | 38.8 |
| MAE | ViT-S/16 | 800 | 38.9 | 35.6 |
| ConMIM[‡] | ViT-S/16 | 300 | **45.8** | **41.0** |
| MAE[‡] | ViT-S/16 | 800 | 41.5 | 37.8 |
| SimMIM[‡] | ViT-S/16 | 800 | 43.0 | 38.7 |
| scratch | ViT-B/16 | - | 46.5 | 41.7 |
| MoCo v3 | ViT-B/16 | 600 | 47.3 | 42.2 |
| BEiT | ViT-B/16 | 800 | 47.4 | 42.1 |
| ConMIM | ViT-B/16 | 800 | 47.8 | 42.5 |
| SimMIM | ViT-B/16 | 800 | 48.7 | 43.2 |
| DINO | ViT-B/16 | 1600 | 47.6 | 42.3 |
| iBOT | ViT-B/16 | 1600 | 48.3 | 42.7 |
| MAE | ViT-B/16 | 1600 | 48.0 | 43.0 |
| BEiT[‡] | ViT-B/16 | 800 | 48.2 | 43.3 |
| SimMIM[‡] | ViT-B/16 | 800 | 48.4 | 43.5 |
| ConMIM[‡] | ViT-B/16 | 800 | **48.7** | **43.6** |
| MAE[‡] | ViT-B/16 | 1600 | 47.8 | 42.9 |

Table 5: Object detection and instance segmentation on COCO (Lin et al., 2014) in terms of AP$^{box}$ (%) and AP$^{mask}$ (%). We tune the optimal learning rate for each model following (He et al., 2022). ([‡]) With intermediate fine-tuning on ImageNet.

tokenizing networks as in BEiT, realizing more efficient and flexible one-stage pre-training. MAE (He et al., 2022) and iBOT (Zhou et al., 2022) cast masked image modeling as reconstruction or distillation tasks rather than vision dictionary look-up. As we discussed before, they require more training epochs for optimal performance as the regularization of regression loss is much more eased than the contrastive loss. Such flaws are especially evident in the performance of small-scale models.

## 5.2 SEMANTIC SEGMENTATION

We evaluate ConMIM on the downstream semantic segmentation using ADE20K (Zhou et al., 2017) benchmark, which consists of 25K images of 150 semantic categories. We use the evaluation metric, mean intersection over union (mIOU), for reference. We use UperNet (Xiao et al., 2018) and adopt the same setup as BEiT (Bao et al., 2022). Images are resized to $512 \times 512$ as input, and the model is fine-tuned for 160K iterations in total. We also use intermediate fine-tuning to fully exploit the potential of the pre-trained models following BEiT (Bao et al., 2022), *i.e.*, first fine-tuning the pre-trained models on ImageNet-1K classification and then transferring to ADE20K. The results are shown in Table 4. We consistently surpass the baseline BEiT with significant improvements, *e.g.*, our 49.8% vs. BEiT's 47.7% on ViT-Base. Moreover, our ConMIM using ViT-Small even achieves comparable performance with BEiT using ViT-Base, *i.e.*, 47.7%. Models pre-trained by masked image modeling generally achieve better performance on segmentation than the ones pre-trained by conventional contrastive learning, *e.g.*, MoCo v3 (Chen* et al., 2021).

## 5.3 OBJECT DETECTION AND INSTANCE SEGMENTATION

For object detection and instance segmentation, we fine-tune Mask R-CNN (He et al., 2017) end-to-end on COCO (Lin et al., 2014). We follow the implementation of (Li et al., 2021) and reproduce all the results in Table 5 since (Li et al., 2021) is still close-source. To tame quadratic complexity with self-attention, most attention blocks in the ViT backbone are replaced with windowed blocks except for four global blocks to perform cross-window interaction. Four up/down-sample modules are evenly placed with ViT to produce pyramid features required by FPN (Lin et al., 2017). The training recipes keep the same with (Li et al., 2021). An input image is resized to $1024 \times 1024$ and augmented by large scale jitter (Ghiasi et al., 2021), with a resize scale of [0.1, 2.0]. We fine-tune the pre-trained models for 25 epochs using an AdamW (Loshchilov & Hutter, 2017) optimizer with a weight decay of 0.1 and cosine learning rate scheduler. All experiments are performed under the same settings, except for the learning rate is specifically tuned for each model, which strictly follows (He et al., 2022) in order to keep fair comparisons as it exploits the potential of each pre-trained method. We achieve much better performance than MAE (He et al., 2022) on ViT-Small.

| Models | ImageNet Acc. |
|---|---|
| DeiT-B (*training from scratch*) | 81.8 |
| MoCo v3 (*conventional contrastive learning*) | 83.2 |
| ConMIM (Ours) | **83.51** |
| denoising patch-level contrast → vanilla instance-level contrast | 82.26 (-1.25%) |
| denoising patch-level contrast → vanilla patch-level contrast | *fail* |

Table 6: Ablation studies on the effect of denoising auto-encoding mechanism.

## 5.4 ABLATION STUDIES

We discuss component design in ConMIM through ablation studies. All the experiments in this section are conducted using ViT-B/16 (Dosovitskiy et al., 2021) pre-trained for 300 epochs, and we report the top-1 classification accuracy after fine-tuning on ImageNet-1K (Deng et al., 2009).

### 5.4.1 ANALYSIS OF DENOISING AUTO-ENCODING MECHANISM

The denoising auto-encoding mechanism is critical to the success of masked image modeling, where the fine-grained patch-level supervision and the masking-and-the-predicting paradigm are the two key factors. We conduct ablation studies in Table 6 to analyze their effect. (1) We remove the entire denoising auto-encoding mechanism from ConMIM, *i.e.*, using average local tokens to perform vanilla instance-level contrastive loss, significant performance drops (-1.25%) are observed. Such a result is even worse than MoCo v3 (Chen* et al., 2021), the state-of-the-art method for conventional contrastive learning. See the visualization in Appendix B.2 for more intuitive comparisons. (2) When only discarding the masking strategy while keeping patch-level contrast, the experiment totally fails due to the trivial information leakage with semantically repetitive patches.

### 5.4.2 ANALYSIS OF PATCH-LEVEL DYNAMIC DICTIONARY

| Models | ImageNet Acc. |
|---|---|
| DeiT-B (*training from scratch*) | 81.8 |
| ConMIM (Ours) | **83.51** |
| *more*: intra-image negative keys → intra-gpu negative keys | 82.60 (-0.91%) |
| *fewer*: intra-image negative keys → filtered intra-image negative keys | 83.12 (-0.39%) |

Table 7: Ablation studies on the dictionary size of ConMIM.

ConMIM performs masked image modeling towards the objective of an intra-image inter-patch contrastive loss, *i.e.*, the dictionary is built individually for each image. The size of the dictionary has effects on the regularization of negative keys. We discuss such a design by either expanding the dictionary or reducing it, as shown in Table 7. (1) We expand the dictionary by gathering other images' keys on the same GPU. Performance drops are observed. The reason might be that the keys from the same image share similar domains and serve as hard negatives for each other. Gathering more negatives from other domains actually ease the regularization. (2) Although patches within the same images can act as hard negatives, they might also be noisy negatives as patches are highly semantically redundant, *e.g.*, patches from different positions may carry similar patterns. So we try to filter them by excluding negative keys that are highly similar to the anchor patch in the latent space. Slight accuracy decreases are shown, indicating that we may exclude some valuable hard negatives. We acknowledge that, although ConMIM can achieve the state-of-the-art results to date, there should be room for further refinement in dictionary design. Further studies are called for.

### 5.4.3 ANALYSIS OF ASYMMETRIC DESIGNS

| Models | ImageNet Acc. |
|---|---|
| DeiT-B (*training from scratch*) | 81.8 |
| ConMIM (Ours) | **83.51** |
| w/o asymmetric image perturbations, use stronger one | 83.41 (-0.10%) |
| w/o asymmetric image perturbations, use basic one | 83.35 (-0.16%) |
| w/ asymmetric image perturbations but switch | 82.62 (-0.89%) |
| w/o asymmetric model progress rates | 81.53 (-1.98%) |

Table 8: Ablation studies on the effect of asymmetric designs.

We introduce two asymmetric designs in ConMIM to enable a stronger denoising auto-encoding mechanism during pre-training. We discuss their effects in Table 8. (1) Asymmetric image perturbations. We use a stronger augmentation for the full input while the basic one for the corrupted image in ConMIM. We try to remove such asymmetric augmentation design and find slight performance drops when using either the stronger one or the basic one for both inputs. More significant performance drops can be observed when directly switching their augmentations. We draw a seemingly counterintuitive conclusion versus conventional contrastive learning. However, it is actually consistent with the theoretical analysis in (Wang et al., 2022b) that the in-training (source) network of contrastive learning requires high variance of input distribution, where the masking strategy already introduces much higher variance than the full input for the momentum (target) network. Further applying stronger augmentation for the corrupted input may lead to a too difficult pretraining task for the network to regularize, similar to the findings in MAE (He et al., 2022). (2) Asymmetric model progress rates. We use a slowly progressing momentum model of the backbone network to embed more challenging but semantically consistent key representations. Such a design is important to avoid information leakage caused by feeding the full input (target) into the same model. When removing it, noticeable accuracy decreases are shown, *i.e.*, -1.98%, even worse than DeiT baseline.

## 5.5 UNCURATED PRE-TRAINING DATA

We would like to see if ConMIM scales well to larger uncurated datasets. Referring to SLIP (Mu et al., 2021), we pre-train on YFCC15M (Thomee et al., 2016), which is about 12 times larger than ImageNet-1k. See the results below in Table 9, ConMIM surpasses both SLIP (using language supervision) and MAE (a comparable pure MIM method) on ViT-B/16, further verifying the effectiveness and scalability of our ConMIM in real-world applications.

| Method | Arch. | #Epochs (YFCC15M) | ImageNet Acc. |
|--------|-------|-------------------|---------------|
| SLIP | ViT-B/16 | 50 | 82.9% |
| MAE | ViT-B/16 | 40 | 83.0% |
| ConMIM | ViT-B/16 | 40 | 83.3% |

Table 9: Scalability on larger uncurated data.

## 5.6 RUNNING TIME

We measure the running time per epoch based on the anchor BEiT (Bao et al., 2022) in Table 10. BEiT requires an additional tokenizer (Ramesh et al., 2021), whose training time is not included here. Note that iBOT (2-view) needs to switch two views and four forward passes in total, and BEiT and ConMIM are 1-view method without switching. MAE is more efficient but achieves inferior performance than our ConMIM. Considering the effectiveness and flexibility, a slightly increasing time over BEiT is acceptable.

| Method | Setup | Time per epoch |
|--------|-------|----------------|
| MAE | 1-view, 1-pass | 0.35× |
| BEiT | 1-view, 2-pass | 1.0× (w/ extra tokenizer) |
| ConMIM | 1-view, 2-pass | 1.05× |
| iBOT | 2-view, 4-pass | 1.40× |
| iBOT (*default*) | (2+10)-view, 14-pass | 2.15× |

Table 10: Running time statistics.

## 6 CONCLUSION AND DISCUSSION

We first propose to perform masked image modeling (MIM) with denoising contrast, a simple and flexible *pure* MIM method with advanced downstream performance. Besides the technical contribution, we would like to emphasize our intriguing insights and potential broader impacts. We are the first to indicate that MIM and contrastive learning both essentially perform vision dictionary look-up and analyze the key factors that make MIM more effective on ViTs, *i.e.*, the fine-grained patch-level constraints and denoising auto-encoding mechanism. More importantly, we hope our work, revitalizing contrastive learning when MIM dominates, would well motivate the community to conduct an objective and comprehensive study of two lines of pre-training methods, and further inspire NLP and multimodal research. Reproducibility is declared in Appendix A, more experimental results in Appendix B, and limitations in Appendix C.

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

# A    REPRODUCIBILITY

We adopt 16 A100 GPUs for pre-training (32 A100 GPUs for ViT-L/16). ViT-S/16 requires less than 1 day for 300 epochs, ViT-B/16 requires around 3 days for 800 epochs and ViT-L/16 requires around 5 days for 800 epochs. A fluctuation within $\pm 0.1\%$ accuracy may be observed on classification when pre-training multiple times with distinct random seeds (use 0 in default).

## A.1    HYPER-PARAMETERS

| Configuration | ViT-S/16 | ViT-B/16 | ViT-L/16 |
|---|---|---|---|
| Layers | 12 | 12 | 24 |
| Hidden size | 384 | 768 | 1024 |
| Attention heads | 6 | 12 | 16 |
| Attention head size | | 64 | |
| Patch size | | $16 \times 16$ | |
| Training epochs | 300 | 800 | 800 |
| Batch size | | 2048 | |
| Adam $\epsilon$ | | 1e-8 | |
| Adam $\beta$ | | (0.9, 0.98) | |
| Peak learning rate | | 5e-4 | |
| Minimal learning rate | | 1e-5 | |
| Learning rate schedule | | Cosine | |
| Warmup epochs | | 10 | |
| Initial momentum coefficient $\alpha$ | | 0.996 | |
| Maximal momentum coefficient $\alpha$ | | 1 | |
| Momentum coefficient schedule | | Cosine | |
| Temperature $\tau$ | | 0.1 | |
| Stoch. depth | 0.1 | None | None |
| Gradient clipping | 3 | 3 | 1 |
| Dropout | | None | |
| Weight decay | | 0.05 | |
| Masking patch size | 16 | 32 | 32 |
| Basic Data Augment | | RandomResizeAndCrop | |
| Strong Data Augment | | ColorJitter(0.4,0.4,0.2,0.1), GaussianBlur(0.1), Solarization(0.2) | |
| Input resolution | | $224 \times 224$ | |

Table 11: Hyper-parameters for ConMIM pre-training on ImageNet-1K.

| Configuration | ViT-S/16 | ViT-B/16 | ViT-L/16 |
|---|---|---|---|
| Peak learning rate | | {1e-3,2e-3,3e-3,4e-3,5e-3} | |
| Batch size | | 1024 | |
| Fine-tuning epochs | 200 | 100 | 50 |
| Warmup epochs | 5 | 20 | 5 |
| Layer-wise learning rate decay | 0.8 | 0.65 | 0.75 |
| Adam $\epsilon$ | | 1e-8 | |
| Adam $\beta$ | | (0.9, 0.999) | |
| Minimal learning rate | | 1e-6 | |
| Learning rate schedule | | Cosine | |
| Repeated Aug | | None | |
| Weight decay | | 0.05 | |
| Label smoothing | | 0.1 | |
| Dropout | | None | |
| Gradient clipping | | None | |
| Stoch. depth | 0.1 | 0.1 | 0.2 |
| Erasing prob. | | 0.25 | |
| Input resolution | | $224 \times 224$ or $384 \times 384$ | |
| Rand Augment | | 9/0.5 | |
| Mixup prob. | | 0.8 | |
| Cutmix prob. | | 1.0 | |
| Color jitter | | 0.4 | |

Table 12: Hyper-parameters for fine-tuning ConMIM on ImageNet-1K classification.

| Configuration | ViT-S/16 | ViT-B/16 | ViT-L/16 |
|---|---|---|---|
| Peaking learning rate | | {1e-5,3e-5,5e-5,7e-5} | |
| Fine-tuning steps | | 160K | |
| Batch size | | 16 | |
| Adam $\epsilon$ | | 1e-8 | |
| Adam $\beta$ | | (0.9, 0.999) | |
| Layer-wise learning rate decay | 0.9 | 0.9 | 0.95 |
| Minimal learning rate | | 0 | |
| Learning rate schedule | | Linear | |
| Warmup steps | | 1500 | |
| Dropout | | None | |
| Stoch. depth | | 0.1 | |
| Weight decay | | 0.05 | |
| Input resolution | | $512 \times 512$ | |
| Position embedding | | Relative | |
| Position embedding interpolate | | Bilinear | |

Table 13: Hyper-parameters for fine-tuning ConMIM on ADE20K semantic segmentation.

| Configuration | ViT-S/16 | ViT-B/16 |
|---|---|---|
| Fine-tuning epochs | | 25 |
| Peaking learning rate | | 1e-4 |
| Learning rate decay | | cosine |
| Adam $\epsilon$ | | 1e-8 |
| Adam $\beta$ | | (0.9, 0.999) |
| Dropout | | None |
| Stoch. depth | | 0.1 |
| Weight decay | | 0.1 |
| Batch size | | 64 |
| Input size | | $1024 \times 1024$ |
| Position embedding | | Abs. + Rel. |
| Augmentation | | LSJ(0.1, 2.0) |

Table 14: Hyper-parameters for fine-tuning ConMIM on COCO object detection and instance segmentation.

## A.2 PSEUDOCODE

---

**Algorithm 1** Pseudocode of ConMIM pre-training in a PyTorch-like style.

---

```
# f: backbone encoder, e.g., vit-base model
# t: temperature, \tau in the paper
# m: momentum, \alpha in the paper

f_slow.params = f.params # initialize
for (x, mask) in loader: # load a mini-batch x with N samples
    # image preprocess with asymmetric perturbations
    x = aug_basic(x) # share same basic aug for paired inputs
    x_full = aug_strong(x)
    x_corrupted = x*(1-mask)+mask_token.expand_as(x)*mask # randomly mask 75% patches

    # build patch-level dynamic dictionaries with asymmetric models
    with torch.no_grad():
        keys = f_slow(x_full) # NxKxD, K is the number of patches
    feats = f(x_corrupted) # NxKxD

    # dictionary look-up with denoising contrastive loss (Eq.(3))
    sim = bmm(feats.view(N,K,D), keys.view(N,D,K)).view(-1,K) # (NxK)xK
    labels = range(K).repeat(N) # (NxK)
    mask = mask.view(-1) # (NxK)
    loss = CrossEntropyLoss(sim[mask]/t, labels[mask])

    # update model
    loss.backward()
    update(f.params)
    f_slow.params = (1-m)*f.params+m*f_slow.params
```

---

bmm: batch matrix multiplication

# B  EXPERIMENTS (CONT.)

## B.1  HYPER-PARAMETER ANALYSIS

| Strategy | Ratio | Acc. |
|----------|-------|------|
| Block | 40% | 82.12 |
| | 60% | 83.20 |
| | 75% | 83.11 |
| | 90% | 82.63 |
| Random | 60% | 82.52 |
| | 75% | **83.51** |
| | 90% | 83.04 |

Table 15: Discuss different masking strategies and ratios.

| temperature $\tau$ | Acc. |
|--------------------|------|
| 0.05 | 82.55 |
| 0.07 | 83.03 |
| 0.1 | **83.51** |
| 0.15 | 83.32 |
| 0.2 | 83.35 |

Table 16: Discuss temperature hyper-parameter in denoising contrastive loss.

| momentum $\alpha$ | Acc. |
|-------------------|------|
| 0.99 | 83.11 |
| 0.996 | **83.51** |
| 0.999 | 83.31 |

Table 17: Discuss momentum coefficient in the slowly progressing model.

**Masking ratio.**   The fine-tuning results on ImageNet using different masking strategies and ratios are shown in Table 15. Randomly masking at a ratio of 75% achieves the optimal performance.

**Temperature.**   We discuss the effect of temperature hyper-parameter in our denoising contrastive loss (Eq. (3)), as illustrated in Table 16. $\tau = 0.1$ achieves the optimal performance empirically. We use this setup for both ViT-Small and ViT-Base.

**Momentum.**   The fine-tuning accuracy on ImageNet using our ConMIM with different values of initial momentum coefficient is shown in Table 17. ConMIM is actually not sensitive to this hyper-parameter and we adopt 0.996 in all experiments for optimal performance.

## B.2  VISUALIZATIONS

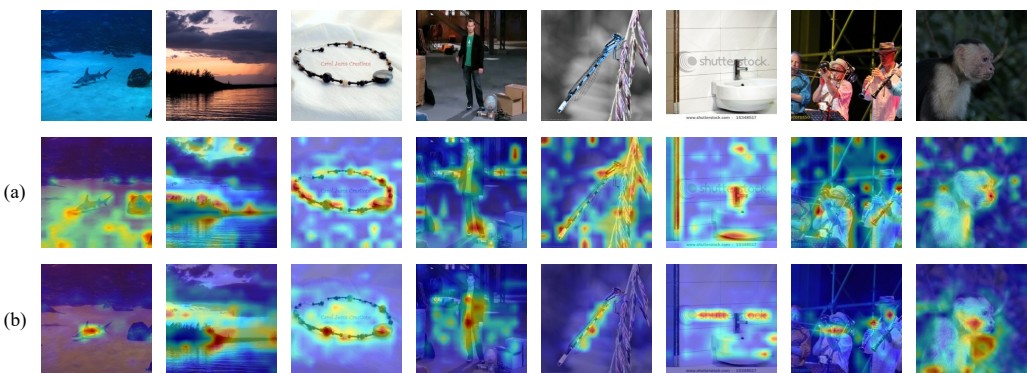

Figure 3: Visualize the self-attention map between [CLS] token and local tokens of the pre-trained ViT-B/16 (Dosovitskiy et al., 2021) model on ImageNet-1K (Deng et al., 2009), where **(a)** indicates ConMIM pretraining and **(b)** indicates the vanilla instance-level contrastive pre-training. Self-attention maps out of 12 attention heads are averaged. It can be observed that ConMIM-pretrained models are much more locally discriminative and aware of the visual context.

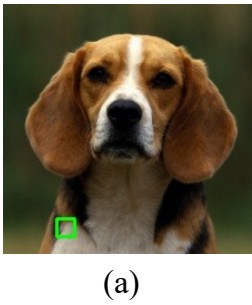 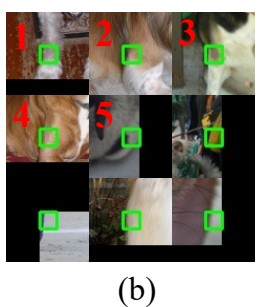 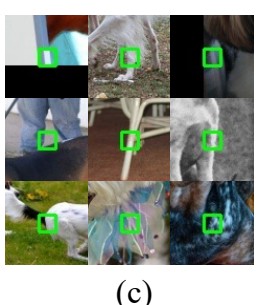

|                  |                  |                  |
| ---------------- | ---------------- | ---------------- |
| (a)              | (b)              | (c)              |

Figure 4: Visualize the dynamic dictionary composed of patches. The dictionary in ConMIM properly provides positive keys with similar semantics while the baseline tokenizer is vulnerable to various low-level changes. **(a)** The query patch from ImageNet validation set. **(b)** Top-ranked patches retrieved by ConMIM-pretrained model. **(c)** Patches out of the same ID (#1813) tokenized by dVAE (Ramesh et al., 2021) in baseline BEiT (Bao et al., 2022).

## B.3 PARTIAL FINE-TUNING

Linear probing is inapplicable to evaluate pure MIM methods which are not designed towards linearly separable instance features. MIM pre-training aims to pursue better pre-trained weights and strong but non-linear features that complement downstream tasks. Linear probing fails to properly measure such properties according to MAE (He et al., 2022) and is also abandoned by BEiT (Bao et al., 2022). For a more comprehensive evaluation, we study partial fine-tuning (a middle ground between linear probing and fully fine-tuning) on ConMIM. As illustrated in Figure 5, fine-tuning a few blocks can achieve accuracy close to full fine-tuning.

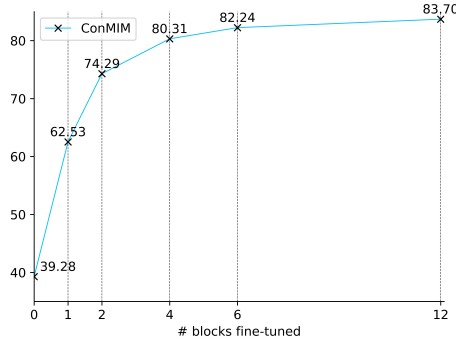

Figure 5: Partial fine-tuning of ConMIM-pretrained ViT-B/16 model.

## B.4 DISCUSSION WITH RELATED WORKS

### B.4.1 COMPARE TO IBOT AND DENSECL

As iBOT (Zhou et al., 2022) is also a tokenizer-free method, someone considers ConMIM a variant of iBOT with DenseCL (Wang et al., 2021) loss. We would like to clarify that our ConMIM **develops more proper constraints for pure MIM** rather than a trivial combination of iBOT and DenseCL, motivated as follows.

Although iBOT abandons the offline tokenizer, it heavily depends on the vanilla DINO loss (*i.e.*, the global self-distillation loss on [CLS] tokens). It actually conducts MIM on top of DINO and fails without the vanilla DINO loss. As an evidence, in Tab. 9 of iBOT's original paper, the fine-tuning accuracy of ViT-S/16 significantly degrades from 81.5% to 79.4% (DeiT-S achieves 79.8%)

| Method | #Views | # Pretrain Epochs | ViT-S/16 | ViT-B/16 |
|--------|--------|-------------------|----------|----------|
| iBOT | 2 | 300 | 81.5 | 82.0 |
| DenseCL | 1 | 300 | 81.4 | 81.9 |
| ConMIM | 1 | 300 | 82.0 | 83.5 |

Table 18: System-level comparison with iBOT (Zhou et al., 2022) and DenseCL (Wang et al., 2021). The results are reported on ImageNet-1K classification in terms of top-1 accuracy (%). iBOT switches 2 standard views for double loss calculating while ConMIM does not.

when removing the vanilla DINO loss. iBOT paper argues that MIM alone hardly captures visual semantics. However, we would like to claim that it is mainly due to the weak patch-level self-distillation loss as MIM constraints in iBOT rather than the issue of pure MIM.

Compared to the regression regularization in self-distillation loss, contrastive learning is proven to be good at structuring the visual space, *e.g.*, MoCo v3 (83.2%, 600ep) beats DINO (82.8%, 1600ep) on ViT-B/16. It is natural to exploit contrastive learning in a pure MIM pretraining method for capturing discriminative visual semantics, but unfortunately, it has never been explored before. We are the first to study it, and introduce a simple and flexible pure MIM method without extra dependencies (*e.g.*, offline tokenizer or global discrimination loss).

DenseCL shows significant differences from our ConMIM in both motivation and method design except for the form of infoNCE (note that infoNCE is widely used for distinct purposes). DenseCL still depends on the global discrimination loss to ensure correct local correspondences and needs to carefully balance the global and local constraints (see Sec. 3.4 of its original paper). Such a chicken-and-egg problem can be seamlessly addressed in our denoising mechanism, including both the masking operation and the asymmetric designs. Moreover, DenseCL hardly encourages the patch-level visual context reasoning as it is a contrastive-only task.

We further provide system-level comparisons in Table 18. For a fair comparison, we retrain iBOT without multi-crop augmentation (but keeping 2 views) using its official code and re-implement DenseCL on ViT architectures. Our ConMIM achieves the optimal performance on both ViT-S/16 and ViT-B/16 architectures.

### B.4.2 COMPARE TO MAE

MAE is a pure MIM method without tokenizer. It casts masked image modeling as a per-pixel denoising reconstruction task with an $\ell_1$ regression loss. The contrastive constraints in ConMIM provide stronger semantic structured regularization than the pixel-level reconstruction loss in MAE, leading to better results with fewer epochs. Moreover, ConMIM is effective especially on small-scale architectures, *e.g.*, ViT-S/16, indicating the necessity of good visual semantic structured constraints on less powerful models. The experimental comparisons are found in Table 19. MAE is trained for 1600 epochs in default and we retrain it using its official code for 800 epochs.

| Method | Arch. | # Pretrain Epochs | Acc. |
|--------|-------|-------------------|------|
| MAE | ViT-S/16 | 800 | 80.9 |
| ConMIM | ViT-S/16 | 300 | 82.0 |
| MAE | ViT-B/16 | 800 | 83.3 |
| ConMIM | ViT-B/16 | 800 | 83.7 |

Table 19: Compare to MAE (He et al., 2022). The results are reported on ImageNet-1K classification in terms of top-1 accuracy (%). Also see the comparison on YFCC15M in Sec. 5.5 of main paper.

### B.5 ANALYSIS OF KEY COMPONENTS

**On the importance of each component.** There are three main components in ConMIM: denoising contrastive loss, asymmetric image perturbations, and asymmetric model progress rates. (a) When changing **denoising contrastive loss** to conventional BEiT loss (classification over token ids), the design of asymmetric image perturbations does not work anymore (see Table 20), and the design of asymmetric model progress rates is inapplicable since the full image encoder in BEiT is a fixed

| Aug. for corrupted image | Aug. for full image | BEiT | ConMIM |
|---|---|---|---|
| weak | weak | 82.9 (*default*) | 83.4 |
| strong | strong | 82.9 | 83.4 |
| strong | weak | 82.7 | 82.6 |
| weak | strong | 82.9 | 83.5 (*default*) |

Table 20: Analysis of asymmetric perturbations using ViT-B/16 pre-trained for 300 epochs. The results are reported on ImageNet-1K classification in terms of top-1 accuracy (%). We observe a similar trend within two methods. Using symmetric perturbations for two views (weak*2 or strong*2) receives identical results within each method, which makes sense as the degree of perturbations for denoising doesn't change. We use asymmetric perturbations to strengthen the denoising mechanism in ConMIM, however, the optimal setup "weak+strong" for ConMIM doesn't work on BEiT since the full image encoder in BEiT is a fixed dVAE, which has certain robustness without training.

dVAE rather than an in-training network. (b) When changing **asymmetric image perturbations** into symmetric ones at the same time keeping the other two components consistent, slight performance drops are found with either strong or weak symmetric perturbations. Though the performance is not optimal, the pretraining is still valid as it outperforms DeiT (training from scratch). See Table 8 of our main paper. (c) When abandoning **asymmetric model progress rates**, the pre-training does not work since the results are even worse than DeiT. The reason might be the shortcut reconstruction with the information leakage. See Table 8 of our main paper.

To conclude, the importance ranking of the three components is **denoising contrastive loss** > **asymmetric model progress rates** > **asymmetric image perturbations**. The denoising contrast is our main method and the asymmetric model progress rates make it works. The asymmetric image perturbations bring extra gains.

**Intuitions of component design.** As analyzed in (Xie et al., 2022), the pretraining task of masked image modeling encourages diverse attention learning in all layers of the Transformer architecture, and introduces locality inductive bias into the patch level. Such properties can also be considered as improving the patch/local uniformity, whereas the contrastive objectives are proven to be good at it (Wang & Isola, 2020). Intuitively, the training objective "per-patch (8192-class) classification in BEiT *vs.* per-patch contrastive in ConMIM" is similar to conventional "1000-class classification in ImageNet supervised pretraining *vs.* instance-level contrastive in self-supervised pretraining". In short, we properly reformulate the MIM task with contrastive objectives to enhance diverse attention in vision Transformers with improved uniformity of patch distributions. As for the asymmetric components, (Wang et al., 2022b) demonstrates that the source network (in-training network updated by loss backpropagation) favors stronger perturbations that introduce high variance into the data distribution. The masking strategy introduces much higher variance than the full input for the momentum (target) network, corroborating the theoretical analysis in (Wang et al., 2022b).

## C  LIMITATIONS

The current version of ConMIM has the following limitations. (1) The intra-image inter-patch contrastive loss may carry some noise as there may exist semantically repetitive patches. (2) We need twice forward operation in each iteration for encoding the full input and corrupted image. If we can reduce such operations to one time, we can save more computation for faster pre-training. (3) ConMIM achieves better performance on downstream detection and segmentation after intermediate fine-tuning. Although it is a common practice in NLP tasks (Devlin et al., 2019), it would still limit the wide range of applications for ConMIM. In future work, we would like to try to address the above limitations, investigate other asymmetric designs for ConMIM, scale up the pre-trained models, and apply ConMIM on multimodal tasks (Ge et al., 2022; Wang et al., 2022a) to exploit its potential in universal representation learning.

