# OpenReview forum: "Masked Image Modeling with Denoising Contrast"
_ICLR.cc/2023/Conference — ICLR 2023 poster_

### Official Review · Reviewer_1DfA · 2022-10-24

**Confidence:** 3
**Clarity, Quality, Novelty And Reproducibility:** 1/ The clarity of this paper is good.…
**Correctness:** 4
**Technical Novelty And Significance:** 3
**Empirical Novelty And Significance:** 3
**Recommendation:** 8

**Strength And Weaknesses:**

Strength:
1/ The proposed ConMIM cleverly combines the conventional contrastive learning and the conventional masked modeling, which gets rid of extra tokenizing networks by revitalizing contrastive learning.
2/ This paper is well-organized and extremely clearly written with some technical contributions.
3/ Figure 1 and 2 are highly easy to understand.
2/ Extensive experiments are conducted to reveal the superiority of the proposed method, in terms of the performance on downstream tasks and running time.
3/ Sufficient ablation studies are conduced to validate the components’ effectiveness (i.e., denoising auto-encoding mechanism, patch level dynamic dictionary, and the asymmetric designs).
4/ Clear contributions are provided.
5/ Detailed experiments settings are provided.

Weakness:
1/ Some aspects of the proposed model seem somewhat to have been empirically defined. Although good experimental results were obtained, it appears that there is no rigorous theoretical background to justify, from a formal perspective, such as the asymmetric image perturbations designs. The validation of the proposed method is mostly from practical evidence.
2/ The results of the proposed ConMIM with more epochs should be provided in Table 1,2,3. For example, in Table 1, only the results of 300 and 800 epochs are provided, but the results of 1600 epochs are missing.
3/ The information about some notations are lack. For example, I haven’t found out what $k$ in Eqn. 1 means so far. Although it is default to represent the keys, I think it would be better to explain it.
4/ Why not alternately augment the corrupted images and full images by strong and weak augment in the pre-training process, instead of augmenting one type of images by a fixed augment (strong or weak augmentation)? This strategy seems more reasonable.


**Summary Of The Paper:**

This paper unleashes the great potential of contrastive learning on denoising autoencoding and introduce a pure MIM method, ConMIM, to produce simple intra-image inter-patch contrastive constraints as the sole learning objectives for masked patch prediction. Additionally, the authors further strengthen the denoising mechanism with asymmetric designs, including image perturbations and model progress rates, to improve the network pre-training.

**Summary Of The Review:**

This paper proposes a novel pre-training method for masked image modeling, namely, ConMIM, to get rid of extra tokenizing networks by revitalizing contrastive learning. Besides, the denoising mechanism with asymmetric designs, including image perturbations and model progress rates are strengthened to further improve the network pre-training. Also, the entire paper is very well-written and the experiments are quite rich.

---

> ### Author Response · Authors · 2022-11-15
> **Responses to Reviewer 1DfA**
>
> Thank you for the valuable and positive comments! We really appreciate the comments for improving the clarity of statements and experimental verifications. The manuscript is revised accordingly, and the main concerns are listed below.
>
> **Q1:** There is no rigorous theoretical background to justify, from a formal perspective, such as the asymmetric image perturbations designs.
>
> **A1:** Thank you for the helpful suggestions that could further enhance our work.
>
> As analyzed in [1], the pretraining task of masked image modeling encourages diverse attention learning in all layers of the Transformer architecture, and introduces locality inductive bias into the patch level. Such properties can also be considered as improving the patch/local uniformity, whereas the contrastive objectives are proven to be good at it [2]. Intuitively, the training objective "per-patch (8192-class) classification in BEiT vs. per-patch contrastive in ConMIM" is similar to conventional "1000-class classification in ImageNet supervised pretraining vs. instance-level contrastive in self-supervised pretraining". In short, we properly reformulate the MIM task with contrastive objectives to enhance diverse attention in vision Transformers with improved uniformity of patch distributions.
>
> As for the asymmetric perturbations, [3] demonstrates that the source network (in-training network updated by loss backpropagation) favors stronger ones that introduce high variance into the data distribution. The masking strategy introduces much higher variance than the full input for the momentum (target) network, corroborating the theoretical analysis in [3]. The design of asymmetric model progress rates is important to avoid information leakage caused by feeding the full input (target) into the same model.
>
> We will conduct a more in-depth analysis with theoretical evidence from a formal perspective in future work. Thank you again for the valuable advice.
>
> [1] Revealing the Dark Secrets of Masked Image Modeling. 2022.
>
> [2] Understanding Contrastive Representation Learning through Alignment and Uniformity on the Hypersphere. ICML 2020.
>
> [3] On the Importance of Asymmetry for Siamese Representation Learning. CVPR 2022.
>
>
>
> **Q2:** Results of more epochs.
>
> **A2:** Thank you for the suggestion. We attempted to scale-up training with more epochs on the ViT-L/16 architecture, achieving better results as shown in the Table below. Due to the limited time in the rebuttal period, we resume from the 800-epoch pre-trained checkpoint for 1600-epoch pre-training and directly reuse the 800-epoch hyper-parameters for fine-tuning and pre-training. Better results are expected to be achieved if training from scratch and specifically tuning for 1600-epoch models. We will try more variants and revise our manuscript with more comprehensive results in the final version.
>
> | Method  | Arch. |   #PT Epochs | ImageNet |
> | ------------- | ------------- | ------------- | ------------- |
> | ConMIM (224)  | ViT-L/16  | 800 | 85.2 |
> | ConMIM (224)  | ViT-L/16  | 1600 | 85.5 (+0.3) |
> | ConMIM (384)  | ViT-L/16  | 800 | 86.3 |
> | ConMIM (384)  | ViT-L/16  | 1600 | 86.5 (+0.2) |
>
>
> **Q3:** Lack of information about notations.
>
> **A3:** Thank you for the suggestion. We have proofread our paper and revised the descriptions accordingly in red.
>
>
> **Q4:** Alternately augment the corrupted images and full images by strong and weak augment in the pre-training process.
>
> **A4:** Thank you for the question. According to your suggestion, we tested such an augmentation paradigm and achieved 83.44% ImageNet classification for ConMIM (ViT-B/16, 800ep, 224), inferior to our original 83.7%. The in-training network regularized by MIM constraints is not suitable for strong augmentation, similar to the findings in the MAE paper [1]. Actually, data augmentation in MIM has been performed with a high proportion (75%) of random masking, which already imposes strong perturbations on the in-training network and introduces a high variance of input. Stronger augmentation on the image, together with random masking, may lead to a too difficult pretext task for the network to regularize.
>
> [1] Masked Autoencoders Are Scalable Vision Learners. CVPR 2022.

---

### Official Review · Reviewer_LV22 · 2022-10-25

**Confidence:** 4
**Correctness:** 3
**Technical Novelty And Significance:** 2
**Empirical Novelty And Significance:** 2
**Recommendation:** 5

**Clarity, Quality, Novelty And Reproducibility:**

This work attempts to combine the two popular approaches, contrastive learning and MIM, for achieving better performance, but it is hard to catch up the key difference with recent approaches that try to use the contrastive learning and MIM.

**Details Of Ethics Concerns:**

N.A.

**Strength And Weaknesses:**

Strengths:
- Quantitative results are promising. The authors validated the proposed method on the models of various sizes, clearly showing the superior performance to prior approaches.
- The method is simple yet effective. It does not leverage extra training stages of image tokenizer.

Weaknesses:
- The authors miss some important reference in the field of MIM. For instance, ‘SimMIM: A Simple Framework for Masked Image Modeling (Z.Xie et al., CVPR 2022)’ was not referenced in this paper. Though SimMIM is remarkably simple in that it applies a simple one-layer prediction head decoder to MIM, it records competitive performance compared to ConMIM. Thus, the effectiveness of the proposed method compared to this simple approach seems rather questionable.

- Similarly, there are various attempts to combine contrastive learning and MIM. For example,

(1) IBOT: Image BERT pre-training with online tokenizer (J.Zhou et al)

(2) Siamese image modeling for self-supervised vision representation learning (C. Tao et al)

(3) Contrastive Masked Autoencoders are Stronger Vision Learners (Z.Huang et al)

The novelty of the proposed method should be clarified with respect to these methods. It was briefly mentioned in the related work section, but it seems that the basic idea itself of the proposed framework (combining the contrastive learning and MIM) is very similar to the above-mentioned methods, and the only difference is what kind of contrastive learning module (or what kind of MIM module) is used in the proposed framework. Namely, it seems that in terms of combining the two approaches, the proposed framework is essentially very similar to the existing methods.

- The asymmetric augmentation used in the proposed method needs more clarification. To the best of my knowledge, applying weak augmentation to the input of the target encoder (slowly progressing ViT in Figure 2) and strong augmentation to the input of the source encoder is known to be suitable for the field of contrastive learning. In contrast, the proposed method contradicts this common wisdom. Do you have any explanation for this?

- Since the proposed method importantly leverages contrastive learning, it seems that the result of combining a variant of contrastive learning approaches (e.g.. DINO, Byol, Simsiam) and MIM should be provided as an ablation study.


**Summary Of The Paper:**

This paper proposes to combine contrastive learning with masked image modeling in order to provide strong pre-text tasks. As opposed to conventional contrastive learning and masked image modeling (MIM) approaches, the authors cast masked image modeling as denoising contrastive learning for an effective vision dictionary look-up. It was validated that the proposed method, ConMIM, competes favorably on downstream tasks (e.g., image classification, semantic segmentation).

**Summary Of The Review:**

The authors proposed a reasonable approach to take the advantage of the two pre-training approaches, but there are some concerns related to novelty, evaluation and method details, as mentioned in ‘Strength and Weaknesses’. Nevertheless, if these comments are well-addressed, the initial rating can be adjusted.

---

> ### Author Response · Authors · 2022-11-15
> **Responses to Reviewer LV22 (1/2)**
>
> Thank you for the constructive comments for improving our clarity of statements and experimental verifications. The manuscript is revised accordingly, and the responses to your main concerns are listed below.
>
> **Q1:** Miss important references, e.g., SimMIM, which records competitive performance.
>
> **A1:** Thank you for the suggestion. We complement the results of SimMIM in our manuscript. As SimMIM paper only reports its ViT-Base performance on ImageNet classification, we reproduce most of the results using its official code, see the table below for details.
>
> | Method  | Arch. |   #PT Epochs | ImageNet | COCO box | COCO mask |
> | ------------- | ------------- | ------------- | ------------- | ------------- |  ------------- |
> | scratch | ViT-Small | - | 79.8 | 43.1 | 38.8 |
> | SimMIM  | ViT-Small | 300 | 79.2 | - | - |
> | SimMIM  | ViT-Small | 800 | 80.1 | 43.0 | 38.7 |
> | ConMIM  | ViT-Small  | 300 | 82.0 | 45.8 | 41.0 |
> | scratch | ViT-Base | - | 81.8 | 46.5 | 41.7 |
> | SimMIM  | ViT-Base | 800 | 83.8(*) | 48.4 | 43.5 |
> | ConMIM  | ViT-Base  | 800 | 83.7 | 48.7 | 43.6 |
>
> (1) SimMIM(ViT-Small, 300ep) performs even worse than random initialization (“scratch”). We attempted to tune better results for it but unfortunately failed. Actually similar conclusions were drawn from the comparison with MAE, see Appendix B.4.2. Such pixel-level regression loss is too weak to properly regularize less powerful models (e.g., ViT-Small) that require good visual semantic structured constraints.
>
> (2) We further transfer the open-source SimMIM(ViT-Base) model on downstream COCO, and it achieves inferior performance than our ConMIM. (*) Also note that, as observed in the issues of SimMIM’s GitHub repo, someone found that the ImageNet classification performance on SimMIM(ViT-Base) can only be reproduced with 16 GPUs, and 0.1~0.2% drops with 8 GPUs (which is the finetuning setup for our ConMIM).
>
> To summarize, our ConMIM is definitely effective, especially on small-size models and fine-grained tasks, further corroborating the conclusions in our paper.
>
>
> **Q2:** In terms of combining contrastive learning and MIM, the proposed framework is similar to [1,2,3].
>
> [1] IBOT: Image BERT pre-training with online tokenizer (J.Zhou et al)
>
> [2] Siamese image modeling for self-supervised vision representation learning (C. Tao et al)
>
> [3] Contrastive Masked Autoencoders are Stronger Vision Learners (Z.Huang et al)
>
> **A2:** Thank you for the question.
>
> (1) We had carefully discussed iBOT in Appendix B.4.1 in the initial version. iBOT can be considered a multi-task framework combining MIM and DINO due to its heavy dependence on the DINO loss (i.e., the global self-distillation loss on [CLS]). Namely, iBOT stacks MIM on the DINO framework. In contrast, ConMIM is a pure MIM method that reformulates the objective of MIM by exploiting the favorable properties of contrastive learning. Please find more details in Appendix B.4.1.
>
> (2) We are sorry for not being aware of [2,3] because they are preprints recently released on arXiv without being officially published. ConMIM is developed independently and concurrently with them. Although we should not be required to compare them according to the ICLR’s policy, we are glad to cite them as concurrent works in our manuscript.
>
> **Q3:** The asymmetric augmentation contradicts the common wisdom in contrastive learning. Any explanations?
>
> **A3:** Thank you for the question. We draw a seemingly counterintuitive conclusion versus conventional contrastive learning, i.e., using weaker perturbation for corrupted input (source network) and stronger perturbation for full input (target network). However, it is actually consistent with the theoretical analysis in [1] that the source network of contrastive learning requires a high variance of the input distribution, where the masking strategy already introduces a much higher variance than the full input for the target network. Further applying stronger augmentation for the corrupted input may lead to a too difficult pretraining task for the network to regularize, similar to the findings in Sec. 4.1 of MAE[2]. We added more detailed explanations in Sec. 5.4.3 of the manuscript.
>
> [1] On the Importance of Asymmetry for Siamese Representation Learning. CVPR 2022.
>
> [2] Masked Autoencoders Are Scalable Vision Learners. CVPR 2022.

---

> > ### Author Response · Authors · 2022-11-15
> > **Responses to Reviewer LV22 (2/2)**
> >
> > **Q4:** The ablation study of combining a variant of contrastive learning approaches (e.g., DINO, Byol, Simsiam) and MIM.
> >
> > **A4:** Thank you for the question. We would like first to recall that we are not aiming to simply combine contrastive and MIM methods, but to reformulate the training objective of masked patch prediction through contrastive learning. Given the non-negative approaches (e.g., BYOL, SimSiam) the reviewer mentioned, we conducted an ablation study that employs the siamese regression objective (i.e., MSELoss) for MIM pretraining. Unfortunately, a much inferior result is achieved, i.e., 81.36% accuracy in terms of ImageNet classification for ViT-B/16 pre-trained for 800 epochs, which is even worse than the random initialization (81.8%).
> >
> > It is not surprising given the similar trends in Tab. 9 of the iBOT paper, where the finetuning accuracy of ViT-S/16 significantly degrades from 81.5% to 79.4% (DeiT-S achieves 79.8%) when using self-distillation loss only for masked patch prediction. All evidence indicates that the patch-level regression loss is too weak to properly constrain the MIM task, probably due to the limited semantics carried by a sole patch that tends to reconstruct patches via shortcuts and leads to trivial solutions if no negatives are provided.

---

> > > ### Author Response · Authors · 2022-11-17
> > > **Looking forward to your reply and the discussions**
> > >
> > > Dear Reviewer LV22,
> > >
> > > Thank you for your comments and valuable suggestions.
> > >
> > > Regarding your main concerns in the initial comments, we (1) added the comparison with SimMIM, (2) discussed iBOT, (3) cited SiameseIM and CMAE in Related Works, (4) explained the asymmetric augmentation, and (5) ablated when using non-negative contrastive objectives.
> > >
> > > We are wondering if our responses and additional experiments have well addressed your concerns. Looking forward to your reply and the discussions!
> > >
> > > Best Regards,
> > >
> > > ICLR 2023 Conference Paper1611 Authors

---

### Official Review · Reviewer_oXdu · 2022-10-25

**Confidence:** 4
**Correctness:** 3
**Technical Novelty And Significance:** 2
**Empirical Novelty And Significance:** 2
**Recommendation:** 6

**Clarity, Quality, Novelty And Reproducibility:**

This paper has described the strong motivation of creating such a backbone Transformer that can better learn image representations to serve applications better. It has made a sufficient literature review with respect to the dictionary look-up, vision transformer, self-supervised learning etc. The originality of the work is good as there is no other similar work.

**Strength And Weaknesses:**

Strength
1. This paper is clearly written. The motivation introduced in the first paragraph and the rich literature reviews are made in the following section.
2. This paper has mainly three contributions, including the dynamic dictionary, contrastive learning as well as asymmetric design.
3. Comprehensive experiments show that the dedicated network based on this model has outperforms the baseline methods consistently on applications including semantic segmentation,  object detection, and instance segmentation.

Weaknesses
1. Regarding the second point of the asymmetric design, why using the plain backbone for corrupted input? The reason of such a design is not clearly explained in the text.
2. Based on Table 8, the asymmetric image perturbation has very little effect on the results.
3. It would be more convincing if the authors would be able to visualize the dynamic dictionary when the training is converged. Make a comparison between this dictionary against other baseline dictionary settings.


**Summary Of The Paper:**

This paper proposes a new masked image modelling (MIM) method, conMIM, to develop better visual representation learning models. The proposed method introduces intra-image inter-patch contrastive learning design to enhance the self-learning network to predict masked patches, which could be quite useful when the training data is limited. This method also develops asymmetric design to promote the training pipeline so that the model is able to learn better. Extensive experiments demonstrate that the proposed backbone can be utilized in many downstream applications well.

**Summary Of The Review:**

In summary, this paper has demonstrated three techniques, patch-level dynamic dictionary and denoising contrastive learning and asymmetric design to enhance a backbone network to learn better image representations. The paper is clearly written and the details and explanations of the key contributions are clearly presented. However, it requires some more in-depth analysis of these techniques beyond the quantitative ablation study. It would be better to introduce the intuitive motivation behind the choice of these techniques.

---

> ### Author Response · Authors · 2022-11-15
> **Responses to Reviewer oXdu (1/2)**
>
> Thank you for the constructive and positive comments! We really appreciate the comments for improving the clarity of statements and analysis. The manuscript is revised accordingly, and the responses to your main concerns are listed below.
>
> **Q1:** Why use the plain backbone for corrupted input?
>
> **A1:** Sorry for the confusion. The term “plain” here refers to the in-training network optimized by loss backpropagation rather than its momentum update version. The in-training network and its momentum update version share the same architectures (e.g., ViT-Small/Base/Large used in our paper), except for different ways to update the model parameters.
>
> The design of asymmetric model progress rates is important to avoid information leakage caused by feeding the full input (target) into the same model. And we need to use the in-training network for corrupted input since the denoising contrastive loss is imposed on the corrupted image rather than the full image that provides denoising targets. We have revised the manuscript accordingly in red for better understanding.
>
>
> **Q2:** The asymmetric image perturbation has very little effect.
>
> **A2:** Thank you for the question. We would like to respond to this question from two perspectives.
>
> (1) *Why study it:* Such a design is motivated by the common practice of establishing positive pairs by two different augmented views of the same instance in conventional contrastive learning. We revitalize contrastive learning in masked image modeling, so it is an intuitive idea to investigate if asymmetric image perturbation is still effective under this setup. See Sec. 5.4.3 of the main paper for our findings and conclusions.
>
> (2) *Why keep it:* We acknowledge that the effect of this component is minimal in our overall framework, as discussed in Appendix B.5. However, we would like to keep it for two reasons. First, as shown in Tab. 8, incorrect use of image perturbations brings significant performance drops (-0.89%). The empirical results provide critical suggestions for follow-up works. Second, the +0.1% improvement over symmetric perturbations on ImageNet is actually not minor. MAE(ViT-B/16, 800ep) achieves +0.1% over BEiT(ViT-B/16, 800ep), as well as iBOT(ViT-B/16, 1600ep, 384) achieves +0.1% over MAE(ViT-B/16, 1600ep, 384).
>
>
> **Q3:** Visualize the dynamic dictionary.
>
> **A3:** Thank you for the suggestion. As the keys of the dynamic dictionary in ConMIM are composed of patches, we perform the visualization with patch retrieval. Specifically, given a query patch from the validation set of ImageNet, we establish its dynamic keys using the other patches and the top-ranked patches serve as positive keys. The keys are sorted according to their similarities with the query patch, where the patch representations are encoded by the pre-trained ConMIM model. As for the baseline BEiT, we sort the keys for the query patch via their discrete IDs encoded by the off-the-shelf tokenizer, i.e., patches out of the same IDs act as positive keys. As illustrated in Fig. 4 of Appendix B.2, we observe that the dictionary in ConMIM properly provides positive keys with similar semantics while the baseline tokenizer is vulnerable to various low-level changes.

---

> > ### Author Response · Authors · 2022-11-15
> > **Responses to Reviewer oXdu (2/2)**
> >
> > **Q4:** More in-depth analysis and intuitive motivation.
> >
> > **A4:** Thank you for the valuable advice.
> >
> > As analyzed in [1], the pretraining task of masked image modeling encourages diverse attention learning in all layers of the Transformer architecture, and introduces locality inductive bias into the patch level. Such properties can also be considered as improving the patch/local uniformity, whereas the contrastive objectives are proven to be good at it [2]. *Intuitively, the training objective "per-patch (8192-class) classification in BEiT vs. per-patch contrastive in ConMIM" is similar to conventional "1000-class classification in ImageNet supervised pretraining vs. instance-level contrastive in self-supervised pretraining".* In short, we properly reformulate the MIM task with contrastive objectives to enhance diverse attention in vision Transformers with improved uniformity of patch distributions. As shown in Appendix B.2 (Fig. 3(a)), visualization of self-attention maps indicates that ConMIM learns locally discriminative patches and well captures the visual context.
> >
> > As for the asymmetric perturbations, [3] demonstrates that the source network (in-training network updated by loss backpropagation) favors stronger ones that introduce high variance into the data distribution. The masking strategy introduces much higher variance than the full input for the momentum (target) network, corroborating the theoretical analysis in [3]. The design of asymmetric model progress rates is important to avoid information leakage caused by feeding the full input (target) into the same model.
> >
> > We will further enhance our work with more in-depth analysis in future work.
> >
> > [1] Revealing the Dark Secrets of Masked Image Modeling. 2022.
> >
> > [2] Understanding Contrastive Representation Learning through Alignment and Uniformity on the Hypersphere. ICML 2020.
> >
> > [3] On the Importance of Asymmetry for Siamese Representation Learning. CVPR 2022.

---

### Decision · Program_Chairs · 2023-01-20

**Decision:**

Accept: poster

**Justification For Why Not Higher Score:**

There are some concerns expressed by the review, and the idea behind the work is not groundbreaking so it may not be up to the oral or spotlight level.

**Justification For Why Not Lower Score:**

This paper has sufficient merits and reviewer support for acceptance so it should not be rejected.

**Metareview: Summary, Strengths And Weaknesses:**

The paper proposes to combine contrastive learning with masked image modeling to create an effective pre-text task. The proposed method, ConMIM, is simple and effective, and performs well on various downstream tasks such as image classification and semantic segmentation. However, there are concerns about the novelty of the method compared to existing approaches that combine contrastive learning and masked image modeling, but the authors addressed the issue the the author response. Overall the paper is well presented, the proposed method represents an interesting approach of combining two families of self-supervised representation learning, so I recommend its acceptance.




**Note From Pc:**

if the above contains the word "oral" or "spotlight" please see: "oral" presentation means -> notable-top-5% and "spotlight" means -> notable-top-25%. As stated in our emails, we are disassociating presentation type from AC recommendations